# Unmet Care Needs and Uncertainty in Patients Newly Diagnosed with Breast Cancer

**DOI:** 10.3390/healthcare10112148

**Published:** 2022-10-28

**Authors:** Li-Wei Chen, Hsu-Huan Chou, Shih-Yung Wang, Whei-Mei Shih

**Affiliations:** 1Department of Nursing, Chang Gung Memorial Hospital, Linko branch, Taoyuan 333, Taiwan; 2Department of General Surgery, Chang Gung Memorial Hospital, Linko branch, Taoyuan 333, Taiwan; 3Graduate Institute of Health Care, Chang Gung University of Science and Technology, Taoyuan 333, Taiwan

**Keywords:** breast cancer, unmet care needs, uncertainty

## Abstract

Breast cancer is the most common cancer in women in the world. Statistics revealed the number of breast cancer patients less than 40 years of age increased over time. Clinical studies found there is a trend of yearly illness happening in younger patients, whose needs related to the illness are different from older patients. The purpose of this study was to explore the uncertainty and needs of patients in different age groups who were newly diagnosed with breast cancer. A descriptive cross-sectional survey study was adopted to recruit 128 patients. The Mishel’s Uncertainty in Illness Scale (MUIS, Chinese version) and the Cancer Needs Questionnaire (short-form, CNQ-SF) were used to collect data. The results revealed that a patient’s age and religion negatively correlated with illness uncertainty. A patient’s age, educational level, work or not, and children’s age correlated with needs for care. Age, illness uncertainty, and educational level were important predictors of the needs of care, with an explanatory power of 29.0%. In Conclusion, patients ≤40 years of age had greater illness uncertainty and needs for care than those who were >40 years of age. For younger patients newly diagnosed with cancer, medical professionals should take the initiative to provide detailed and complete information on breast cancer treatment plans, prognosis, and home self-care, which helps clarify the possible future treatments and results and further enables patients to acquire self-care skills and knowledge.

## 1. Introduction

According to the World Cancer Research Fund International, breast cancer is the most common cancer in women in the world, and the number of breast cancer deaths ranks fifth among cancer deaths worldwide [1]. In 2018, of the nearly 8.8 million newly identified cancer cases, breast cancer accounted for 12.3% of the total cases, and contributing 25.4% of the cases were women. According to a cancer registration report from Taiwan, the ranking of female breast cancer mortality in 2019 was the third of the top ten causes of death, with 22.1 per 100,000 populations [2]. The crude incidence rate was 148.85%, ranking first in women. In the 2019 registration data from Taiwan, the peak age group of breast cancer occurred in 15.5% between 45 and 49 years old, followed by 14.27% in 60–64, 14.18% in 55–59, 14.04% in 50–54, 11.12% in 65–69, and 9.86% in 40–44 years of age group and the median age was 56 years old. In clinical care, we found that younger breast cancer patients may not only have to face the pain of surgical treatment, the impairment of the body’s mental integrity, and the threat of death, but also the problems of marriage, childbirth, care of children and the elderly at home, and work, etc. It can be seen that they are exhausted and full of unknown fear and anxiety about the future, which leads to the exploration of the needs of young breast cancer patients. Following the accurate breast tumors analysis, the most popular treatments of breast cancer, including surgical intervention, therapeutic radiation, cytotoxic chemotherapies, and molecularly targeted agents [3,4,5], led to patient illness uncertainty and needs of care, especially for a patient newly diagnosed with breast cancer. Therefore, it is crucial to understand the views of unmet care needs and the uncertainty of disease on patients newly diagnosed with breast cancer when facing, many abovementioned events for the first time.

Many studies have found that common needs for cancer patients include information on treatment, symptom control, medical information or counseling, psychological support, communication with healthcare systems, daily care, assistance for returning to society, resource connect, substance support, and psychological emotion and spiritual support [6,7,8,9]. In the psychosocial system review, breast cancer threatens patient’s self-integrity and life for women, and the destruction of body image causes hair loss and breast and weight changes. Young patients are looking for a standard image after breast surgery and seek breast reconstruction more often than older. The diagnosis and treatment of breast cancer affect the relationship between husband and wife, as well as with children and older parents, especially during treatment. Family relationships are significant for cancer patients because they provide a high level of social support, and younger women are more likely to be strained than older people in intimate relationships. Additionally, young patients have more loneliness and a stronger sense of isolation from their peers and support systems [10]. Based on the above studies, the needs of cancer patients cover many areas, including physical, mental, and even social needs, especially in young patients.

Mishel [11] suggested that the feeling of uncertainty is an important factor affecting the patient’s experience of illness and hospitalization. With a lack of disease-related information, patients are not aware of the hospital routine, the effectiveness of the treatment, or treatment side effects. In addition, patients oftentimes cannot have clear and detailed communication with doctors or nurses, resulting in great uncertainty. The studies by Chien, Huang, Han, Kuo, and Yu [12] and Lee, Shaw, Sheu, Chang, and Hsu [13] suggested that the degree of uncertainty in breast cancer patients was moderate. Sajjadi [14] revealed that cancer patients’ uncertainty had a significant positive correlation with age and a significant negative correlation with education. In Wu, Lin, Li, and Jian’s [15] study on gynaecological cancer patients indicated that the average rating of unclear experience was higher than the complex feeling and work been affected after the illness. The study by Lin, Chiang, Acquaye, Vera-Bolanos, Gilbert, and Armstrong [16] showed that uncertainty and the five negative mood states were directly associated with symptom severity perceived by patients (*p*  <  0.01 for all). Hsiang, Lee, Li, and Cheng [17], studied the uncertainty in female breast cancer patients and found that years of education negatively correlated with illness uncertainty and age positively correlated with illness uncertainty. Thus, the above literature indicates that patient uncertainty is associated with personal factors, disease characteristics, and the degree of distress caused by physical symptoms.

The needs of patients during the course of illness affect their quality of life, and therefore, in addition to their physical functional status, the physical, mental, and social impacts, as well as their feelings in daily life, are all important evaluation points. Clinical studies found there is a trend of yearly illness happening in younger patients, whose needs related to the illness, physiology, and information are oftentimes different from older patients. When diagnosed with breast cancer, patients may feel that their future is unpredictable, that their lives are threatened, and uncertain. In a state of uncertainty, patients may search for relevant information to reduce the negative impact of the illness. The feeling of uncertainty interferes with the patient’s ability to search for information related to illness [13]. Therefore, the objectives of this study were to explore the uncertainty and cancer-related needs in patients newly diagnosed with breast cancer and compare the differences in uncertainty and cancer-related needs and their influencing factors between different age groups.

In summary, the hypotheses of this study, according to the literature review, were the significance of illness uncertainty and the needs for care for younger patients.

## 2. Materials and Methods

### 2.1. Design and Sample

A cross-sectional design was conducted to collect 128 patients newly diagnosed with breast cancer to explore cancer-related needs and their uncertainty. Study tools included the demographic data, the Cancer Needs Questionnaire-short-form (CNQ-SF, Chinese version), and the Mishel Uncertainty in Illness Scale (MUIS, Chinese version).

This study was conducted in a surgical ward of a medical center in northern Taiwan. Patients were divided into two groups based on their age: 40 years was used as a cut-off according to the literature review [18,19] and the American Cancer Society [20]. One group was less than or equal to 40 years of age, and the other group was greater than 40 years of age. The number of samples estimated by G-power using 0.8 as satisfied power, Effect size 0.3 for moderate effect, α is 0.05 for significance level, plus the expected loss rate of 10% requires 128 patients. All participants met the inclusion criteria: (1) initial diagnosis of breast cancer and waiting for surgery; (2) at least 18 years or older; (3) conscious and able to communicate verbally; and (4) consent for an interview. The exclusion criteria included patients not diagnosed with breast cancer for the first time, who could not speak in the Mandarin or Taiwanese language, or who refused to participate in this study. 

### 2.2. Study Tools

Demographic data such as age, education, occupation, children’s age, and religion were collected for analysis.

CNQ-SF was developed by Lattimore-Foot [21] and mainly measures the status of needs in cancer patients, assessing the patient’s needs in multiple areas. This scale included five subscales: psychological, health information, physical and daily living, patient care and support, and interpersonal communication. Chen, Lai, Cheng, Liao, and Chang [22] translated the CNQ-SF into Chinese. The Cronbach’s α value for internal consistency reliability was 0.97. It is a 5-point Likert scale with “1” indicating “no needs or not applicable” to “5” indicating a high need to assess the degree of unmet care needs. The higher the score, the higher the degree of unmet care needs.

MUIS was developed by Mishel in 1981 to mainly measure patient uncertainty about symptoms, diagnosis, treatment, and the relationship with the caregiver. The Chinese version was translated and validated by Dr. Sheu [23], with Cronbach’s α value for internal consistency reliability being 0.87. On the scale, 15 questions were for assessing ambiguity, and 10 questions were for assessing complexity. A 5-point Likert Scale was used for scoring, with “1” point indicating “strongly disagree” and “5” points indicating “strongly agree”, totaling from 25 to 125. The higher the score, the higher the uncertainty.

### 2.3. Data Collection and Statistical Analysis

Prior to data collection, to ensure the rights and interests of the patients, the study plan was first approved by the Human Rights Review Committee of the hospital. Data were collected from patients who were newly diagnosed with breast cancer and were waiting for surgery by the researcher. The purpose of the questionnaire, methods, and the time required were explained to the patient, and patient consent was acquired. Data were processed, and statistical analyses were performed using the SPSS/Windows 22 software package for percentage and mean for descriptive analysis; *t*-test and ANOVA were used for the difference of demographic data in both illness uncertainty and unmet care needs, the correlation for both illness uncertainty and unmet care needs, and regression for predicting factors affecting both illness uncertainty and unmet care needs.

## 3. Results

### 3.1. Basic Attributes

A total of 128 patients were recruited for this study. The results showed that 79.7% (102 patients) were older than 40 years and 20.3% (26 patients) were less than 40 years of age. The mean age was 49.4 ± 11.34 years. Eighty-two percent of them were married. Most (83, 64.8%) of them had high school (vocational school) education and above. Nearly 56% of them had occupations. More than half of them (54.7%) had 1–2 children. Seventy-two percent of patients had religion. Forty-eight percent of them were responsible for the source of income. Eighty-nine percent of them lived with their family members. Eighty-two of them had no family history of the illness. Most (58.6%) had a spouse as the main caregiver. Most (73.43%) of them had surgical resection as the treatment method.

The total mean score of CNQ-SF was 86.27 (ranging between 32 and 151), and the mean score for a single question was 2.70. The mean score for “health information needs” (M = 3.95) was highest, followed by “psychological needs” (M = 2.60), “patient care and support needs” (M = 2.25), and “interpersonal communication needs” (M = 2.09); “physical and daily living needs” (M = 2.05) had the lowest score. The above results showed that only the mean score for a signal question under “health information needs” was nearly 4 points, indicating that subjects had very high unmet care needs in this area.

The mean illness uncertainty score was 62.23 ± 11.25 (between 25 and 95), and the mean score for a single question was 2.49 (out of 5), which was below 3 points, suggesting that patients felt uncertain about illness, although not strong. The score for illness ambiguity was higher (M = 2.66), and that for illness complexity was lower (M = 2.22).

As shown in Table 1, there were significant differences in “age” in overall unmet care needs (*p* < 0.05) as well as in physical and daily living needs (*p* < 0.05), psychological needs (*p* < 0.01), and interpersonal communication needs (*p* < 0.05) except for health information needs, suggesting that patients with ≤40 years of age had greater overall unmet care needs than those who were >40 years old. As for “education”, “occupation”, and “children’s age”, there were significant differences in overall unmet care needs (*p* < 0.05, *p* < 0.01, and *p* < 0.001) as well as in physical and daily living needs (*p* < 0.05, *p* < 0.01, and *p* < 0.05), psychological needs (*p* < 0.05, *p* < 0.01, and *p* < 0.01), interpersonal communication needs (*p* < 0.05, *p* < 0.01, and *p* < 0.01), and patient care and support needs (*p* < 0.05, *p* < 0.01, and *p* < 0.01) expect for health information needs. Patients with junior college and above had higher physical and daily living needs, psychological needs, interpersonal communication needs, patient care and support needs, and overall unmet care needs than patients with junior high school and below. Patients with occupation had greater overall unmet care needs than those without work. Patients with younger age of children tended to have greater overall unmet care needs.

As shown in Table 2, there was significant differences in “age” in overall illness uncertainty (*p* < 0.001) as well as in illness ambiguity (*p* < 0.01) and illness complexity (*p* < 0.001). Patients ≤ 40 years of age had a significantly greater perception of overall illness uncertainty than those who were > 40 years old. As for “religion”, there were significant differences in overall illness uncertainty (*p* < 0.05) and illness ambiguity (*p* < 0.05). Patients without religion had a significantly stronger perception of overall illness uncertainty than those who had a religion.

### 3.2. Correlation between Basic Attributes, Unmet Care Needs, and Illness Uncertainty

As shown in Table 3, age (r = −0.264, *p* < 0.01), educational level (r = 0.316, *p* < 0.01), occupation (r = −0.276, *p* < 0.01), and children’s age (r = 0.326, *p* < 0.01) were correlated with overall unmet care needs in terms of physical and daily living, psychological, interpersonal communication, and patient care and support except for health information. The younger age of patients, the greater of overall unmet care needs they needed. The higher the educational level, the greater of overall unmet care needs they needed. As for occupation, patients with occupation tended to have greater overall unmet care needs. Patients with younger age of children had greater overall unmet care needs.

As shown in Table 4, age was negatively correlated with overall illness uncertainty (r = −0.393, *p* < 0.01), illness ambiguity (r = −0.384, *p* < 0.01), and disease complexity (r = −0.265, *p* < 0.05) suggesting that patients with younger age had higher perception of overall illness uncertainty. Religion was negatively correlated with overall illness uncertainty (r = −0.181, *p* < 0.05) and illness ambiguity (r = −0.208, *p* < 0.05), suggesting that participants without religion had a higher perception of overall illness uncertainty.

As shown in Table 5, Overall illness uncertainty was positively correlated with overall unmet care needs (r = 0.483, *p* < 0.01). The results suggested that the stronger the perception of overall illness uncertainty in patients, the greater their unmet care needs in all areas.

### 3.3. Influencing Factors of the Overall Unmet Care Needs

A stepwise regression method was used to identify influencing factors, including patient age, religion, and illness uncertainty. The illness uncertainty was the dependent variable, and the *p* value of the overall F-test for the final multiple regression model was significant (F = 22.970, *p* < 0.001), which suggested that the explanatory power of the regression model was statistically significant (r^2^ = 0.154) and that the independent variable (age) in the model could jointly effectively affect the dependent variable “illness uncertainty”, with the explanatory power of 14.7% (Table 6). Results of *t*-test analysis showed that the correlation coefficient between “illness uncertainty” and “age” was significant and negative (β = −0.393, *p* < 0.05), suggesting the older the patient, the lower the degree of perception of illness uncertainty.

When “unmet care needs” was a dependent variable, the *p* value of the overall F-test for the final multiple regression model was significant (F = 12.139, *p* < 0.05), which suggested that the explanatory power of the regression model was statistically significant (R2 = 0.290), and that the independent variables (illness uncertainty, educational level) in the model could jointly effectively affect the dependent variable “unmet care needs”, with the explanatory power of 29.0% (Table 7). Results of *t*-test analysis showed that the correlation coefficient between “illness uncertainty” and “educational level” was significant and positive (β = 0.262, *p* < 0.05), indicating that the greater the uncertainty in patients, the greater the unmet care needs; and the higher the educational level, the greater the unmet care needs.

## 4. Discussion

The results of this study revealed that the younger the age of breast cancer patients, the more unmet needs and uncertainty of disease they had. 

### 4.1. Correlation between Basic Attributes of Breast Cancer Patients and Their Needs for Care

Analysis of unmet care needs in breast cancer patients showed that the mean score for “overall unmet care needs” was 86.27, with a mean score of 2.70 for a single question. Among all areas, the mean score for “health information” (M = 3.95) was the highest, followed by “psychological” (M = 2.60) and “patient care and support” (M = 2.25), which is similar to the results of the survey by Patterson et al. and Tsai [24,25] on the needs of cancer patients and their family members. The mean score for “health information” was nearly 4, suggesting that subjects had great needs in this area. In the CNQ-SF, “wish to be informed of the test results as soon as possible” (M = 4.14, SD = 1.20) had the highest score, indicating that subjects had the highest level of needs in this area, followed by “needs information that helps restore health”, “needs information that helps improve treatment effectiveness”, and “needs adequate information on the success rate of the treatment”. Burg et al.’s study [26] showed breast cancer survivors identified more unmet needs than other survivors. The results suggested that when caring for breast cancer patients, in addition to attending to the patients’ needs for illness perception, it is even more necessary to promptly provide information on medical treatments. When necessary, the healthcare team members should explain the patient’s health condition and treatment in detail to clarify any concerns about future treatments.

The analysis of the relationship between subject basic attributes and unmet care needs found that differences were significant by subject “age”, “educational level”, “occupation”, and “children’s age” (*p* < 0.05). The patient’s age and educational level significantly negatively correlated with “physical and daily living”, “psychological”, “interpersonal communication”, “patient care and support”, and “overall unmet care needs”, suggesting that when patients were younger or had an occupation, the levels of their needs in these areas were higher. Patients’ educational level and the age of their children significantly positively correlated with their “physical and daily living”, “psychological”, “interpersonal communication”, “patient care and support”, and the “overall unmet care needs”, suggesting that when patients had higher levels of education or when their children were ≤ 20 years old, the levels of their needs in these areas were higher. The results were in line with the findings by Burg et al. [26] that older cancer survivors identified fewer unmet needs on average than younger survivors as well as findings by Sarkar et al. [27] that report patients reported the highest unmet supportive care needs in the domain health system and information followed by psychological needs. The higher the self-care ability of the patient, the greater their needs, which may be due to the fact that when patients had a higher degree of autonomy, they had greater needs related to their illness. Since younger patients have more needs for care, nurses should take into consideration younger groups in using communication skills to engage them in conversation, providing counseling, and taking the initiative to provide nursing-related instructions during patients’ daily nursing activities to strengthen patients’ ability to respond to problems.

### 4.2. Correlation between Basic Attributes of Breast Cancer Patients and Their Illness Uncertainty

The average score of disease uncertainty, in this case, was 62.23 points (SD = 11.25) (between 25 and 95), indicating patients suffering from medium–high disease uncertainty. With regards to illness uncertainty, the mean score for a single question under “illness ambiguity” (M = 2.66) was higher, indicating subjects felt a moderate level of illness uncertainty, in line with the results of previous studies on uncertainty in female breast cancer patients [2,12,13,14,15,17,28,29] In a qualitative study, women with newly diagnosed breast cancer experience intense uncertainty about the disease, treatment, and social relationships and thus exhibit avoidance behavior [30].

When assessing illness uncertainty in patients, the top three questions receiving high scores were “I am not sure the degree of discomfort I may feel”, followed by “I do not know what may happen next”, and “because of the treatment, the things I can or cannot do are always changing”, which were the same as the top three questions identified in a previous study on the relationship between illness uncertainty and depression in gynaecologic cancer patients, although the order was slightly different. In addition, the results were slightly different from the findings by Ko et al. [31] and Lee et al. [13] concerning the uncertainty in breast cancer patients in terms of the order of the top three questions. The results were completely different from the top three items identified by Hsiang et al. [17], which were conducted on 100 women who were newly diagnosed with stage 1 to stage 3 breast cancer and underwent a mastectomy in the absence of metastasis. The reasons may be that subjects in this study first received outpatient diagnosis and confirmation and then were admitted to the hospital for surgery and further treatment. At the outpatient clinic, physicians explained the illness condition and treatment plans, and during the course of treatment, patients took the initiative to discuss relevant information found on the internet, and therefore patients had a better understanding of the disease and the treatment process. However, it is unable to know about cancer grading until the pathological tissue reports have been confirmed. Patients felt that many things were changing; they were not able to predict the disease progression and therefore felt uncertain about the future. Studies found that patient age, marital status, educational level, and religious beliefs correlated with illness and affected condition after diagnosis for work uncertainty, in line with the results of various studies on uncertainty [13,15,17,31,32,33].

Patient age and religious beliefs negatively correlated with illness uncertainty, suggesting that patients ≤ 40 years of age had greater illness uncertainty than those who were >40, and that patients who did not have religious beliefs had greater illness uncertainty than those who had a religion. Patients with religious support tended to lower uncertainty [34]. The result that younger patients had greater uncertainty was different from the findings by Hsiang et al. [17] and Hagen et al. [28]. The reason may be that the percentage of career women was higher than before, and their socioeconomic status also plays an important role. During the clinical practice and interaction with the patients, older patients expressed they had nothing to fear from their experience. Just do what the doctor said. However, younger patients, since admitted to the hospital, often repeatedly asked about treatment plans, treatment methods, prognosis, worrying about leaving children at home, and whether their work would be affected. Therefore, it is recommended that for younger patients newly diagnosed with breast cancer, nurses should provide information related to breast cancer care as soon as they are admitted to the unit, and the nursing instructions on various treatment methods, possible side effects, and home self-care precautions, should be strengthened in order to increase their knowledge of the disease and reduce younger patient uncertainty. In the meantime, social workers and counselors were also available for these patient groups to consult.

### 4.3. Correlation between Patients’ Illness Uncertainty and Needs of Care

Analysis of the relationship between patients’ illness uncertainty and unmet care needs found that illness ambiguity and overall illness uncertainty significantly positively correlated with physical and daily living, interpersonal communication, patient care and support, health information, and overall unmet care needs. The results suggested that when patients had a higher perception of illness uncertainty, the levels of their unmet care needs in all areas were also higher, in congruence with the results by Mishel [35]. Nurses caring for patients newly diagnosed with breast cancer should pay attention to both illness uncertainty and the needs for care.

### 4.4. Influencing Factors of the Overall Unmet Care Needs

A stepwise regression analysis indicated that age could negatively affect illness uncertainty, with an explanatory power of 14.7% (β = −0.393, *p* < 0.001).

The *t*-test results showed that the correlation coefficient between illness uncertainty and age was significant and negative (β = −0.393, *p* < 0.05), suggesting that when patients were older, the levels of their perception of illness uncertainty were lower.

When “unmet care needs” was the dependent variable, the *p* value of the overall F-test for the final multiple regression model was significant (F = 12.139, *p* < 0.05), indicating the explanatory power of the regression model was significant (r^2^ = 0.290), and that independent variables (illness uncertainty, educational level) in the model could jointly effectively affect the dependent variable “unmet care needs”, with the explanatory power of 29.0%. The *t*-test results showed that the correlation coefficient between illness uncertainty and educational level was significant and positive (β = 0.262, *p* < 0.05), suggesting that when patients had greater illness uncertainty, their unmet care needs were also greater. Furthermore, when patients had higher levels of education, their unmet care needs also increased, similar to part of the results by Lee et al. [13], Liao et al. [33], and Kim et al. [31].

### 4.5. Study Limitations and Suggestions

Subjects in this study were recruited from a medical center in northern Taiwan and cannot be generalized. During patient recruitment, young cancer patients were often sadder or felt that it was an unsuitable time and refused to participate in the study. Therefore, there were fewer young patients included in this study, and the results of this study may not be able to be applied to all breast cancer cases in Taiwan. Future studies should expand the study area and increase the sample size in order to increase study inference and universality.

## 5. Conclusions

This study found that patient age was an important predictor of uncertainty. The younger age of patients, the greater illness uncertainty and unmet care needs they had. Illness uncertainty and educational level were important predictors of unmet care needs. Healthcare professionals should enhance professional knowledge and skills in observation and communication, have a more in-depth understanding of the source of illness uncertainty, evaluate patient needs, and guide patients to express their feelings.

## Figures and Tables

**Table 1 healthcare-10-02148-t001:** Differences in unmet care needs between subjects’ basic attributes (*n* = 128).

Subject Basic Attribute	Number of Subjects	Physical and Daily Living	Psychological	Interpersonal Communication	Patient Care and Support	Health Information	Total
		*M* ± *SD*	*M* ± *SD*	*M* ± *SD*	*M* ± *SD*	*M* ± *SD*	*M* ± *SD*
Age							
(1) ≤40	26	14.77 ± 5.76	35.77 ± 11.62	5.08 ± 2.47	14.73 ± 6.21	26.96 ± 8.50	97.31 ± 28.71
(2) >40	102	11.66 ± 5.55	26.80 ± 13.74	3.99 ± 2.39	13.21 ± 6.91	27.63 ± 8.38	83.28 ± 30.87
*t* value		2.53	3.06	2.06	1.02	−0.361	2.096
*p* value		0.013 *	0.003 **	0.042 *	0.308	0.719	0.038 *
Education							
(1) Junior high school and below	45	10.82 ± 5.20	22.91 ± 12.00	3.53 ± 2.22	10.87 ± 5.58	26.00 ± 8.60	74.13 ± 27.39
(2) High (vocational) school	38	12.08 ± 5.19	29.11 ± 14.50	4.13 ± 2.24	14.39 ± 7.24	28.34 ± 8.41	88.05 ± 30.56
(3) Junior college and above	45	13.98 ± 5.99	33.96 ± 12.21	4.89 ± 2.45	15.42 ± 6.32	28.64 ± 7.78	96.89 ± 28.97
*F* value		3.766	8.33	3.79	6.15	1.36	7.07
*p* value		0.026 *	0.004 **	0.025 *	0.028 *	0.262	0.012 *
Scheffé test		3 > 1	3 > 1	3 > 1	2 > 1, 3 > 1		3 > 1
Occupation							
(1) Yes	71	13.52 ± 5.68	31.83 ± 12.86	4.69 ± 2.59	15.03 ± 6.80	28.65 ± 7.77	93.72 ± 29.78
(2) No	57	10.79 ± 5.17	24.65 ± 13.51	3.56 ± 1.96	11.63 ± 6.09	26.35 ± 8.79	76.98 ± 28.47
*t* value		2.81	3.07	2.80	2.94	1.57	3.22
*p* value		0.006 **	0.003 **	0.006 **	0.004 **	0.120	0.002 **
Children’s age							
Not applicable	21	13.05 ± 4.43	32.33 ± 10.31	4.38 ± 2.29	14.52 ± 6.86	27.24 ±7.74	91.52 ± 26.25
(1) 1–10 years	17	15.29 ± 6.37	33.94 ± 11.68	4.88 ± 2.57	15.29 ± 4.82	27.12 ± 7.13	96.53 ± 26.09
(2) 11–20 years	32	13.75 ± 5.92	35.97 ± 14.43	5.09 ± 2.41	16.16 ± 7.42	29.72 ± 8.19	100.69 ± 31.39
(3) 20 years and above	58	10.36 ± 4.98	21.69 ± 11.38	3.41 ± 2.15	11.17 ± 5.98	26.76 ± 8.82	73.40 ± 27.29
*F* value		5.24	12.04	4.41	5.04	1.93	7.88
*p* value		0.019 *	<0.0001 ***	0.006 **	0.003 **	0.430	<0.0001 ***
Scheffé test		2 > 4, 3 > 4	1 > 4, 2 > 4, 3 > 4	3 > 4	3 > 4		2 > 4, 3 > 4

Note: * *p* < 0.05, ** *p* < 0.01, *** *p* < 0.001.

**Table 2 healthcare-10-02148-t002:** Differences in illness uncertainty between subject basic attributes (n = 128).

Subject Basic Attribute	Number of People	Illness Ambiguity	Illness Complexity	Overall Illness Uncertainty
		*M* ± *SD*	*M* ± *SD*	*M* ± *SD*
Age				
(1) ≤ 40	26	44.88 ± 10.10	24.15 ± 4.10	69.04 ± 13.17
(2) > 40	102	38.68 ± 7.97	21.63 ± 3.74	60.50 ± 10.01
*t* value		3.35	3.02	3.71
*p* value		<0.001 ***	0.003 *	<0.001 ***
Religion				
(1) No religion	36	42.97± 9.22	22.50 ± 3.77	65.47 ± 12.28
(2) Has religion	92	38.92 ± 8.39	22.04 ± 4.03	60.97 ± 10.62
*t* value		2.39	0.59	2.06
*p* value		0.018 *	0.557	0.041 *

Note: * *p* < 0.05, *** *p* < 0.001.

**Table 3 healthcare-10-02148-t003:** Correlation between subjects’ basic attributes and their unmet care needs.

Subject Basic Attribute	Physical and Daily Living	Psychological	Interpersonal Communication	Patient Care and Support	Health Information	Total
Age	−0.254 **	−0.320 **	−0.208 *	−0.186 *	−0.058	−0.264 **
Educational level	0.237 **	0.342 **	0.239 **	0.287 **	0.134	0.316 **
Occupation	−0.243 **	−0.264 **	−0.236 **	−0.254 **	−0.138	−0.276 **
Children’s age	0.225 *	0.373 **	0.268 **	0.273 **	0.130	0.326 **

* *p* < 0.05, ** *p* < 0.01.

**Table 4 healthcare-10-02148-t004:** Correlation between subjects’ basic attributes and illness uncertainty.

Subject Basic Attribute	Illness Uncertainty	Illness Complexity	Overall Illness Uncertainty
Age	−0.384 **	−0.265 *	−0.393 **
Religion	−0.208 *	−0.052	−0.181 *

* *p* < 0.05, ** *p* < 0.01.

**Table 5 healthcare-10-02148-t005:** Correlation coefficients between patient illness uncertainty and unmet care needs in different areas.

Needs for Care	Illness Uncertainty		
	Illness Ambiguity	Illness Complexity	Overall Illness Uncertainty
Physical and daily living	0.392 **	0.146	0.357 **
Psychological	0.507 **	0.262 **	0.487 **
Interpersonal communication	0.420 **	0.206 *	0.400 **
Patient care and support	0.410 **	0.184 *	0.384 **
Health information	0.373 **	0.021	0.299 **
Total	0.526 **	0.207 *	0.483 **

* *p* < 0.05, ** *p* < 0.01.

**Table 6 healthcare-10-02148-t006:** Stepwise regression analysis of illness uncertainty in breast cancer patients.

Input Variable	r	r^2^	Δr^2^	*F*	Standardized Coefficient β	*t*	*p*
Illness uncertainty							
Age	0.393	0.154	0.147	22.970	−0.393	−4.793	0.000

Note: r, multivariate correlation coefficient; r^2^, decisive coefficient; Δr^2^, change in decisive coefficient; β, standardized regression coefficient.

**Table 7 healthcare-10-02148-t007:** Stepwise regression analysis of unmet care needs in breast cancer patients.

Input Variable	r	r^2^	Δr^2^	*F*	Standardized Coefficient β	*T*	*p*
Unmet care needs							
Illness uncertainty	0.483	0.233	0.227	38.351	0.452	5.997	0.000
Educational level	0.549	0.301	0.290	12.139	0.262	3.484	0.001

Note: r, multivariate correlation coefficient; r^2^, decisive coefficient; Δr^2^, change in decisive coefficient; β, standardized regression coefficient.

## Data Availability

The datasets used and/or analyzed during the current study are available from the corresponding author upon reasonable request.

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
