# Peer review of "Unmet Care Needs and Uncertainty in Patients Newly Diagnosed with Breast Cancer"

_healthcare, 2022, doi:10.3390/healthcare10112148_

Round 1
Reviewer 1 Report
Authors have performed a cross sectional study to know the needs and uncertainty of breast cancer patients. This is a nice study with interesting results, but I have some comments to adrees before this paper to be published:
*In the introduction, lines 45 to 47 does not fit with the previous idea you expose. I would suggest to better link these ideas; is it not well explained how the treatment may affect cancer patients.
*In lines 66 to 83, I'm not sure if this information is really needed on the introduction. I wuold try to synthesize the ideas exposed on these articles. It increases the length of the introduction and, in my opinion, makes the reader not to focus on the objective of your study, but the results and conclusions of others'
*It would be interesting to add an hypothese at the end of your introduction.
Methods section:
* Although sample size is calculated, which outcome did you use to calculate it?
* Was there any exclusion criteria?
*Please add also what other outcomes did you register on your patients (e.g. Education, ocupation, children age... and so on)
*Could you please explain with more details the statistical analysis? This would help readers understand it. Moreover, late in the results you perform a correlation, but this is not mentioned on this section.
*Line 148: why do you explain again the CNQ? it is already explained on methods. If you want to add some more information about this questionnaire, please do it on the methods section.
Discusion:
First paragraph: I don't get why the first information is about diet when you do not expose anything about this on the text.
One of the main findings of the regression analysis is the religion, however you do not seem to describe this a lot neither include it on the conclusion... in my opinion this is an important finding to discuss.
*Line 343: there should be a mistake at the end of this line. "Younger patients" does not link with the following sentence
*Line 365: you did not start the conclusion in another line. Please check carefully the composition of your manuscript to verify that everything is on its place.
Reviewer 2 Report
Please look for the typing mistakes. the conclusion should be in line 366.
Give more detailed information about the statistical analysis.
The number of subjects included in different age group and also together is not adequate. I would suggest the authors to focus on one age group rather then younger or adult patients together.
Reviewer 3 Report
It is a nice paper. Breast tumor segmentation is an important steps for diagnosing breast cancers. Some papers related to breast tumours analysis could be cited, such as:
Zhou, L., Wang, S., Sun, K., Zhou, T., Yan, F., & Shen, D. (2022). Three-dimensional affinity learning based multi-branch ensemble network for breast tumor segmentation in MRI. Pattern Recognition, 129, 108723.
